

# Thermal treatment and leaching of biochar alleviates plant growth inhibition from mobile organic compounds

Nigel V. Gale, Tara E. Sackett and Sean C. Thomas

Faculty of Forestry, University of Toronto, Toronto, Ontario, Canada

## ABSTRACT

Recent meta-analyses of plant responses to biochar boast positive average effects of between 10 and 40%. Plant responses, however, vary greatly across systems, and null or negative biochar effects are increasingly reported. The mechanisms responsible for such responses remain unclear. In a glasshouse experiment we tested the effects of three forestry residue wood biochars, applied at five dosages (0, 5, 10, 20, and 50 t/ha) to a temperate forest drystic cambisol as direct surface applications and as complete soil mixes on the herbaceous pioneers *Lolium multiflorum* and *Trifolium repens*. Null and negative effects of biochar on growth were found in most cases. One potential cause for null and negative plant responses to biochar is plant exposure to mobile compounds produced during pyrolysis that leach or evolve following additions of biochars to soil. In a second glasshouse experiment we examined the effects of simple leaching and heating techniques to ameliorate potentially phytotoxic effects of volatile and leachable compounds released from biochar. We used Solid Phase Microextraction (SPME)–gas chromatography– mass spectrometry (GC-MS) to qualitatively describe organic compounds in both biochar (through headspace extraction), and in the water leachates (through direct injection). Convection heating and water leaching of biochar prior to application alleviated growth inhibition. Additionally, growth was inhibited when filtrate from water-leached biochar was applied following germination. SPME-GC-MS detected primarily short-chained carboxylic acids and phenolics in both the leachates and solid chars, with relatively high concentrations of several known phytotoxic compounds including acetic acid, butyric acid, 2,4-di-tert-butylphenol and benzoic acid. We speculate that variable plant responses to phytotoxic organic compounds leached from biochars may largely explain negative plant growth responses and also account for strongly species-specific patterns of plant responses to biochar amendments in short-term experiments.

## INTRODUCTION

"Biochar" is the term given to charcoal intended for use as a soil amendment, commonly derived from pyrolysis of biomass residues from forestry and agriculture. Biochar has been recently heralded for its ability to increase productivity and ameliorate poor soil

Corresponding author
Nigel V. Gale,
nigel.gale@mail.utoronto.ca

conditions, while mitigating anthropogenic climate change by enhancing soil carbon sequestration (*Lehmann, 2007*; *Biederman & Harpole, 2013*). Many biochars exhibit high surface areas, pH, and cation exchange capacity (CEC), properties that can increase soil fertility, nutrient availability, and water retention (*Major et al., 2010*; *Atkinson, Fitzgerald & Hipps, 2010*). Meta-analyses of field and greenhouse studies suggest average increases of 10–30% in aboveground biomass in response to applications of biochar for agricultural crops (*Jeffery et al., 2011*; *Liu et al., 2013*; *Biederman & Harpole, 2013*), and larger increases averaging ~40% for trees (*Thomas & Gale, 2015*). Plant responses, however, vary greatly across species and ecosystems, and null or negative biochar effects are common (*Jeffery et al., 2011*; *Spokas et al., 2012*). Thus, null and negative responses of plants to biochar have received recent research attention (*Buss & Mašek, 2014*; *Kołtowski & Oleszczuk, 2015*; *Domene et al., 2015*; *Buss et al., 2015*).

Large positive plant responses to biochar are commonly observed in tropical and boreal soils, with much lower responses commonly reported in temperate soils (*Jeffery et al., 2011*; *Thomas & Gale, 2015*). Strong positive plant responses to biochar in infertile, acidic tropical soils are likely mainly due to a combination of soluble P from biochar and P retention by biochar (*Mukherjee & Zimmerman, 2013*), sorption of salts or metals by biochar (*Park et al., 2011*; *Lashari et al., 2013*), and soil liming effects (*Major et al., 2010*). Recent results by *Pluchon et al. (2014)* suggest a positive relationship between P in wood-derived biochar and growth responses of tree seedlings, indicating possible P limitation in boreal soils. The sorption of phenolic compounds by biochar has also been suggested to contribute to high plant responses to biochar in some boreal soils (*Wardle, Zackrisson & Nilsson, 1998*). Increased soil basicity from strongly basic biochars can limit the availability of Ca, Mg, P and other nutrients (*Major et al., 2010*; *Makoto et al., 2011*; *Marks, Alcañiz & Domene, 2014*), and such responses are expected to be most pronounced on soils that are already neutral to basic (*Kloss et al., 2014*). Biochar's affinity for cations and sorption of anions can also "lock up" certain soil nutrients, in particular, mineralized N (*Glaser, Lehmann & Zech, 2002*; *Asai et al., 2009*).

A rather unexplored explanation for null and negative plant responses to biochar is related to plant exposure to volatile compounds generated during pyrolysis and re-condensed as liquids on biochar's surface or trapped as gases within pore spaces (*Spokas, Baker & Reicosky, 2010*; *Spokas et al., 2011*; *Hale et al., 2012*). Pyrolysis converts the polymeric constituents lignin, cellulose, and hemicellulose (as well as starches, lipids, and "extractives" such as terpenoid and phenolic compounds) into liquid bio-oil, charcoal, and non-condensable gases (*Shafizadeh, 1982*). Many of the compounds generated are "mobile" (*Buss & Mašek, 2014*; *Buss et al., 2015*): they are either water soluble or volatile, i.e., have high vapor pressures under ambient temperature and pressure conditions (*Yu et al., 2007*). Such compounds include low molecular weight alcohols, ketones, aliphatic acids, and phenols (*Buss et al., 2015*; *Lievens et al., 2015*), as well as larger polyaromatic hydrocarbons (PAHs) (*Hale et al., 2012*; *Kołtowski & Oleszczuk, 2015*; *Domene et al., 2015*). Other mobile and potentially toxic compounds produced during pyrolysis such as salts and heavy metals are generally detected at only low levels in biochars derived from non-contaminated feedstocks (*Domene et al., 2015*),

are strongly sorbed by biochar (*Thomas et al., 2013*; *Lucchini et al., 2014*), and are therefore less likely to be an important component of toxicity responses to biochars (*Buss & Mašek, 2014*; *Kołtowski & Oleszczuk, 2015*; *Domene et al., 2015*; *Lievens et al., 2015*).

The quantities and types of mobile organic compounds present in biochar will depend on the composition of the feedstock and the conditions of pyrolysis. Biomass feedstocks differ greatly in the composition of cellulose, hemicellulose, and lignin and generate a diversity of volatile compounds. Woody biomass generally has higher concentrations of lignin and cellulose (20–25% and ~45%, respectively) than non-woody grasses (which typically have lignin concentrations of 3–12%: *Morrison, 1972*). There is, however, remarkably high variation in these constituents among woody species (*Pettersen, 1984*). Thermal decomposition of cellulose occurs around 400 °C and releases the highest proportion of labile, aliphatic compounds. Higher pyrolysis temperatures (> 400 °C), where decomposition of lignin dominates, produce syn-gases containing higher concentrations of carbon dioxide ($CO_2$), ethylene ($C_2H_4$), and ethane ($C_2H_6$) (*Bilbao, Millera & Arauzo, 1989*; *Kloss et al., 2012*). Re-condensation of these volatiles from cold spots due to poor insulation and blockage of syn- or pyrolysis gases, and their capture within biochar pores, has been suggested in a number of studies (*Spokas et al., 2011*; *Buss & Mašek, 2014*). Mobile organic compounds have been characterized in a number of wood-derived biochar's produced at moderate pyrolysis temperatures 350–600 °C (*Spokas et al., 2011*; *Buss et al., 2015*; *Lievens et al., 2015*) and have demonstrated toxicity in a variety of organisms (*Buss & Mašek, 2014*; *Kołtowski & Oleszczuk, 2015*). Fast pyrolysis and gasification biochars have been suggested to exhibit the greatest phytotoxic effects (*Rogovska et al., 2012*).

Some mobile organic compounds released from biochar in soils are hormetic, i.e. stimulating plant growth at low dosages but inhibiting growth at high dosages (*Baldwin et al., 2006*; *Calabrese, Iavicoli & Calabrese, 2012*). The aqueous fraction of the bio-oil, commonly referred to as "wood vinegar," contains primarily acetic acid and phenolic compounds. When diluted ($10^3$–$10^7$ times), wood vinegars commonly stimulate plant growth (*Mu, Uehara & Furuno, 2003*; *Mu, Uehara & Furuno, 2004*), but remain potent insecticides and fungicides (*Yatagai et al., 2002*; *Velmurugan, Han & Lee, 2009*). At higher concentrations these compounds can be herbicidal. *Buss & Mašek (2014)* found germination inhibition in cress from leached compounds from acidic chars (pH = 3.64), attributed to low molecular weight organic acids and phenols (*Buss et al., 2015*). Because ethylene is a gaseous plant hormone involved in growth regulation, plant stress signalling, and tissue senescence (*Abeles, Morgan & Saltveit, 1992*; *Ortega-Martínez et al., 2007*), its evolution from freshly produced biochars (*Fulton et al., 2013*) is another plausible mechanism for reduced growth responses. Ethylene evolution from biochars would also be expected to result in pronounced changes in plant growth form and reproduction (*Abeles, Morgan & Saltveit, 1992*). To our knowledge phytotoxicity from mobile organic compounds released by biochar has not been specifically tested in plant growth beyond the germination stage, or in plants grown in soil media other than pure sand.

To test the effects of biochar derived from forestry residues on temperate plant performance in a managed system, we grew *Lolium multiflorum* and *Trifolium repens* in a

factorial greenhouse experiment. We examined effects of addition rate (0–50 t/ha), biochar type (three different biochar produced from sawdust), and application method (top-dressing vs complete soil mixes). We hypothesized that plant growth would increase overall, and would vary with biochar type, dosage, and application method. Because of exclusively negative and null effects observed in the first experiment, in a second greenhouse experiment, we examined the effects of simple post-treatment biochar processing techniques to ameliorate potentially toxic effects of mobile organic compounds released by biochars. Because volatiles sorbed by biochars during pyrolysis exist in liquid (re-condensed) and gaseous phases (*Spokas, Baker & Reicosky, 2010*; *Spokas et al., 2011*; *Hale et al., 2012*), and can be water-leached (*Jonker & Koelmans, 2002*; *Hale et al., 2012*; *Buss et al., 2015*; *Lievens et al., 2015*), we tested both biochar leaching and heating treatments on the growth responses of *Lolium multiflorum*. We hypothesized that water leaching and convection heating would significantly improve plant responses by reducing volatile and leachable compounds. We predicted more pronounced growth enhancement with biochars that were leached longer and heated at higher temperatures. We also hypothesized that application of biochar leachate would significantly inhibit plant growth. To identify candidate molecules involved in toxicity responses, we examined volatile and leachable organic compounds using semi-quantitative Solid-Phase-Micro-Extraction (SPME)–gas chromatography–mass spectrometry (GC-MS) of solid chars and water leachates.

## MATERIALS AND METHODS

### Study species

The herbaceous species *Lolium multiflorum* Lam. (annual ryegrass: hereafter ryegrass) and *Trifolium repens* L. (white clover: hereafter clover) are commonly used as temperate forage crops, for erosion control (*Ledgard & Steele, 1992*), and are early colonizers of temperate sites following fire events (*Keeley et al., 1981*; *Milberg, 1995*). Ryegrass is a fast growing, nutrient-demanding annual dicot that can tolerate a range of soil conditions. Clover is a slow-growing perennial monocot that has low nutrient requirements as its associated root symbionts can biologically fix atmospheric nitrogen (*Ledgard & Steele, 1992*). Clover in previous studies has shown increased nitrogen fixation in soils amended with biochar (*Quilliam, DeLuca & Jones, 2013*). Because of this contrast in life histories, resource use, and growth forms, we expected that changes to soil properties would have differential effects on growth and performance in these two species. Because larger effects of biochar on growth have been reported for annuals over perennials (*Biederman & Harpole, 2013*), and since the strongest negative effects in experiment 1 were detected in *L. multiflorum*, it was selected to test if biochar post-treatments could enhance plant responses in experiment 2.

### Experiment 1: effects of amount, type, and application method

Plants were grown in 11.33 cm$^2$ surface area growth containers (Ray Leach SC-10 cone-tainers; Stuewe and Sons, Tangent, OR, USA) 21 cm depth, 164 mL volume, with a fiberglass screen placed at the base of the container to reduce soil loss in soil amended with

**Table 1 Physical and chemical properties of biochars.** Properties of maple and spruce sawdust biochar made in an 80 L batch pyrolyzer with highest treatment temperature between 350–450 °C (from *Sackett et al. (2015)*), and maple/birch sawdust biochar produced in a flow-through screw fed pyrolyzer with highest treatment temperature of 500–600 °C. Three replicates were used for each characterization.

| Attribute | Maple flow through BC | Maple batch BC | Spruce batch BC |
|---|---|---|---|
| Moisture (%) | 2.7 (0.0) | 2.43 (0.15) | 2.4 (0.100) |
| Ash (%) | 2.74 (0.06) | 1.67 (0.06) | 0.65 (0.06) |
| Volatile matter (%) | 29.87 (3.07) | 29.64 (2.64) | 21.38 (0.32) |
| Fixed carbon (%) | 64.68 (3.07) | 66.25 (2.64) | 75.57 (0.32) |
| pH | 7.39 (0.23) | 7.7 (0.50) | 6.5 (0.0) |
| EC ($\mu$S/cm) | 105 (4) | 75 (10) | 72 (14) |
| *Elemental composition* | | | |
| C (%) | 85.01 (2.3) | 77.3 (3) | 77.9 |
| N (%) | 0.71 | 0.10 (0.01) | 0.01 |
| S (%) | n/a | 0.02 (0.0) | 0.01 |
| Phosphorous (mg/kg) | n/a | 197 (16) | 75 |
| Ca (mg/kg) | n/a | 6,015 (944) | 2,419 |
| K (mg/kg) | n/a | 3,443 (105) | 1,243 |
| Mg (mg/kg) | n/a | 619 (6) | 285 |
| *Particle size distribution* | | | |
| > 4,000 $\mu$m (%) | 0.76 (0.95) | 0.40 (0.11) | 3.09 (1.91) |
| 2,000–4,000 $\mu$m (%) | 15.37 (1.52) | 5.61 (0.91) | 17.39 (0.98) |
| 500–2,000 $\mu$m (%) | 57.25 (2.63) | 31.53 (2.54) | 59.14 (0.89) |
| 250–500 $\mu$m (%) | 13.49 (0.84) | 27.74 (2.43) | 11.44 (0.227) |
| 125–250 $\mu$m (%) | 9.18 (2.42) | 25.32 (0.74) | 7.05 (0.59) |
| 125–63 $\mu$m (%) | 3.63 (1.48) | 9.02 (0.36) | 1.85 (0.31) |
| < 63 $\mu$m (%) | 0.30 (0.15) | 0.39 (0.16) | 0.01 (0.02) |

**Note:**
Values not determined are abbreviated as n/a, not applicable.

three sawdust-derived biochars (Table 1): 1) sugar maple sawdust pyrolyzed in a batch unit at 475 °C (MB) (~1.5 h total pyrolysis time), 2) spruce sawdust pyrolyzed in a batch unit at 422 °C (SB) (~1.5 h total pyrolysis time), and 3) 80% sugar maple, 20% yellow birch sawdust pyrolyzed in a flow-through pyrolysis unit at 575 °C (MFT) (~0.5 h total pyrolysis time). The batch unit was purged with nitrogen gas, and the flow-through unit consisted of a long feed-screw that resulted in tight packing of feedstock to limit oxygen access during pyrolysis. Biochars were applied at 5, 10, 20, and 50 t/ha either mixed into the soil or applied to the surface; the control treatment consisted of soil without biochar. These biochars were selected for the present study as they are expected to be used in industrial scale applications and their effects on temperate plant and soil functioning have been the subject of recent study (*Sackett et al., 2015*; *Noyce et al., 2015*; *Noyce et al., 2016*, *Mitchell et al., 2015*). Five replicates were used for the biochar treatments and 10 replicates for were used for the control treatments (to increase statistical power for control vs treatment contrasts). In total, 4 addition rates (5, 10, 20, and 50 t/ha) × 3 biochars (MB, SB, MFT) × 2 application methods (mixed, top-dressed)

**Table 2 Soil physical and chemical properties.** Properties of the drystic cambisol soil used in both experiments collected from the uppermost mineral layer in a temperate forest, Haliburton, ON. Soil analysis was done at the Agriculture and Food Laboratory, University of Guelph, Guelph, ON. Test procedures and units are reported in parentheses. Three replicates were used for all analyses.

| Attribute | Value |
| --- | --- |
| Moisture (%) | 2.72 |
| Organic matter (%) | 12.3 |
| $NH_4$ (KCl-$NH_4$, mg/kg) | 3.8 |
| $NO_3$-N (KCl-$NH_4$, mg/kg) | 148 |
| P ($NaHCO_3$, mg/L) | 32 |
| Mg ($NH_4C_2H_3O_2$, mg/L) | 140 |
| K ($NH_4C_2H_3O_2$, mg/L) | 62 |
| pH ($CaCl_2$) | 6.8 |

× 5 replicates), + (10 control plants (no biochar, field soil) × 2 species (clover, ryegrass), resulted in n = 260 plants. Biochar use is expected to target acidic, coarse-textured soils of low nutrient status. Accordingly, the soil used in the experiment was a drystric cambisolic field soil (or drystric brunisol based on the *Canadian System of Soil Classification (1998)*), that was sandy-loam in texture (Table 2), collected in the summer of 2011 from the uppermost mineral layer (< 10 cm) of a managed forest at Haliburton Forest and Wildlife Reserve, Haliburton, ON, Canada (45°15′N, 78°34′W), and re-used from control treatments in a previous experiment. All plants were supplemented with 0.082 g of slow-release NPK fertilizer representing relatively high inputs found in disturbed temperate herbaceous systems (*Garbutt & Bazzaz, 1987*): addition rates are 130 kg/ha (N), 5 kg/ha (P), 48 kg/ha (K), respectively. Germination was initiated by placing seeds on greenhouse benches in water-saturated vermiculite. One seedling was transplanted into each prepared treatment pot five days after germination. Fertilizer was applied following planting. A randomized block design was employed using five blocks in which plants were randomly placed, with each treatment combination was represented once in each block. Supplemental lighting was applied to maintain a 16 h photoperiod. The experimental growth period was 28 days (December 20, 2012–January 17, 2013) and 41 days (December 20, 2012–January 30, 2013) for ryegrass and clover, respectively. Average daily greenhouse temperature throughout the experiment was 16.5 °C with average daily maximums and minimums of 20 and 16 °C, respectively. All plants were top watered semi-daily (daily to every other day) to saturation and allowed to drain.

## Experiment 2: modulation of effects through leaching and heating treatments

To test the effect of pre-application biochar heating and leaching on plant performance, ryegrass was grown in pots (as above) with additions of MFT biochar processed to remove mobile compounds before additions. Because, there was no difference in plant performance between biochars in experiment 1, suggesting similar compounds responsible for the observed phytotoxicity, only MFT was selected for experiment 2 since it was most available at the time of experimental set-up. Two leaching treatments and three heating

treatments were applied to biochars before being applied as both mixed and top-dressings at a dose of 10 t/ha (n = 8). We leached biochar with 1:1 (v:v) mixture of biochar to de-ionized water for either 0.5 h (WW-0.5) or 24 h (WW-24) on an oscillating table. Biochar-water slurries were then suction filtered using Whatman #4 filter paper, and the filtrate captured. Filtrate from the leachings was collected and 5.0 mL of filtrate was applied as a separate treatment to plants (n = 8) one day following planting (Leach-0.5, Leach-24). Leached biochars were then rinsed with 400 mL of de-ionized water, suction filtered, and applied to their respective treatments. Biochars were placed in a convection oven to be heat treated for 24 h at 50 (Heat-50), 100 (Heat-100), and 150 °C (Heat-150), cooled, and added to soil. To assess the effect of leaching and heating, un-processed biochar treatments (mixed and top-dressing) were also established (n = 8), as well as a control treatment containing no biochar (n = 10). In total, 2 leaching treatments + 3 heating treatments + 1 un-processed biochar × 2 application methods × 8 replicates + 2 leachate additions × 8 replicates + 10 controls, resulted in n = 122 plants. Soil used in the experiment was the same field soil as used in experiment 1 (Table 2). Experimental growth period was 28 days (January 28–February 26, 2013). Average daily greenhouse temperature was 18 °C, with daily average maximum and minimum temperatures of 21 and 15 °C, respectively. All plants were top watered semi-daily to saturation.

## Biochar and soil characterization

Biochar was characterized (Table 1) as follows: electrical conductivity (EC) and pH were measured in a 1:20 (w:v) biochar to $H_2O$ solution using pH and EC probes (*Rajkovich et al., 2012*). Particle size distribution was determined from dry sieving following the ASTM D2862 method (*ASTM, 1999*). Total moisture content, ash content, volatile matter, and organic matter content were determined by oven drying at 105 °C, by muffle furnace combustion at 750 °C, by heating at 950 °C in sealed containers for 7 min, and through loss on ignition for 0.5 h in a muffle furnace at 500 °C by following ASTMD1762-84 (*ASTM, 2007*). Total carbon and nitrogen were quantified using combustion analysis using an Elementar VarioMax (Elementar Analysensteme GmbH, Hanau, Germany). Total P and cations ($K^+$, $Ca^+$, and $Mg^{2+}$) for MB and SB were obtained using a sulphuric acid digest and Mehlich III extraction, respectively, then analyzed by ICP-OES (SPECTRO Analytical Instruments GmbH, Kleve, Germany) (see *Sackett et al., 2015*).

Soil characterization (Table 2) was conducted at the Agriculture and Food Laboratory, University of Guelph, Guelph, ON.

## Plant growth measurements

For both experiments, above- and below-ground plant biomass and leaf area were measured at the end of the experiments. Because leaf area indicates plant carbon gain, nutrient uptake, and physiological performance, and has demonstrated responsiveness to biochar additions (*Graber et al., 2010*), we measured leaf area using a Li-3100C leaf area scanner with a resolution of 1 $mm^2$ (LiCor Biosciences, Lincoln, Nebraska, USA). Below-ground biomass was separated from soil and biochar media by dry sieving followed by gentle washing of roots with water. Above-ground and below-ground biomass, including scanned leaves, was
dried for 48 h at 65 °C and weighed. Since increased nutrient uptake, nitrogenase activity, and nodulation has been reported in legumes amended with biochar (*Rondon et al., 2007*; *Quilliam, DeLuca & Jones, 2013*), before drying we counted nodules on clover roots under a dissection microscope. Only nodules with leghemoglobin were scored (visible red colour).

## SPME–GC-MS of volatile and leachable compounds

We followed *Rombolà et al. (2015)* for qualitative analysis of SB and MB solid char and their water leachates, with minor modifications. Briefly, for headspace (HS)-SPME of solid biochars, 0.5 grams of sample was placed in a 10 mL headspace vial and spiked with 2.0 mL of 1 ppm *O*-eugenol as an internal standard. HS vials were placed on a heating plate at 150 °C for 30 min while a Carboxen-PDMS fiber was inserted into the top 1 mL of the headspace vial on a Agilent 7890A GC system. Following heating the fiber was inserted into the injector and analytes thermally desorbed at 250 °C for 10 min prior to direct injection. GC analysis was carried out using a DB-WAX column (20 m length, 0.20 μm width, 0.10 mm i.d) following the thermal program in *Rombolà et al. (2015)*: 80 °C for 5 min, then 10 °C/min to 250 °C for a total run time of 28 min. MS analysis was carried out on a 5975C Agilent MS with peak identification conducted using the NIST reference spectral library. Detection was made under electron ionization at 60 m/z and an acquisition of 1 scan/s. Hits with over 70% probability and relative match factors higher than 800 were considered; peaks of internal standards were confirmed.

Direct injection (DI)-SPME of leachates was performed on 3 mL of dionized water leachates (following washing method used for MFT) with 1.0 mL of 2 M $KH_2PO_4$ buffer. This sample was spiked with 2.5 mL of 1 ppm *O*-eugenol, and 2.5 mL of 5 ppm 2-ethyl butyric acid as internal standards, and then placed into 10 mL headspace vials. The Carboxen-PDMS fiber was directly inserted into the solution under magnetic stirring for 30 min at 250 °C. GC-MS analysis was performed as above. Quantification of analytes was not attempted due to the un-reliability of SPME in producing accurate calibration curves with internal standards, a problem also noted in *Spokas et al. (2011)*. All GC-MS analysis were conducted at the Teaching and Research in Analytical Chemistry and Environmental Sciences (TRACES) Centre, University of Toronto, Scarborough (1065 Military Trail, Toronto, ON, M1C 14A). SPME-GC-MS was conducted on SB and MB only since MFT was no longer available at the time of quantification.

## Statistical analysis

Both experiments had treatment structures that consist partially of a full set of factorial combinations, have multiple controls, and test other treatments of interest (i.e. have an augmented factorial design); we therefore used multiple contrasts to test our hypotheses (following *Schaarschmidt & Vaas, 2009*). This allows the pair-wise comparison of both individual treatments (e.g., MFT biochars vs control) and groups of treatments (e.g, all biochars at 5 t/ha vs control). The test statistic for each contrast was calculated from:

$$T = (\Sigma c_i \hat{u}_i)/(\sigma \Sigma c_i / n_i)$$

Where $\hat{u}_i$ is the mean estimate derived from the linear model described in *Bretz, Hothorn & Westfall (2002)*, and $c_i$ is the contrast coefficient, and $\sigma$ represents the residual error. Contrast coefficients are shown in Tables S1–S3. Simultaneous confidence intervals (95%) were derived from:

$$\left[ \Sigma c_i \hat{u}_i \pm (q_1 - \alpha, \, M, \, R) \, \sigma \left( \sqrt{\Sigma c_i^2 / n_i} \right) \right]$$

Where the critical value, $(q_1 - \alpha, \, M, \, R)$, is calculated from the multivariate t-distribution with dimension $M$ and correlation matrix $R$ (see *Bretz, Genz & Hothorn, 2001*) for critical value determination. Simultaneous confidence intervals were used to display the magnitude and direction of mean comparisons. Analyses were done using the "multcomp" package (*Bretz, Hothorn & Westfall, 2002*; *Hothorn et al., 2008*) in the statistical software R version 3.1.0 (*R Core Team, 2012*). Many biochar studies have complex treatment structures and treatments of interest: we expect wide use of multiple contrast tests in future biochar research as an effective and informative analysis tool, we have therefore included the R script as a supplementary file for use as a reference.

To test the effect of biochar addition rate on plant performance traits we performed linear regressions with above- and below-ground biomass and leaf area as response variables and with biochar dosage as the independent variable. We log-transformed response variables to represent effect sizes metrics reported in recent meta-analyses of plant growth responses to biochar (*Liu et al., 2013*; *Biederman & Harpole, 2013*; *Thomas & Gale, 2015*) following: RR = ln (B/C), where RR is the response ratio metric, B is the mean biomass of biochar treated plants, and C is the mean biomass of control plants without biochar. Regressions were performed for each of the three biochars used in the experiment and were not separated by application method (top-dressing vs complete mixtures). Analyses were conducted in the statistical software R version 3.1.0 (*R Core Team, 2012*).

## RESULTS

### Experiment 1: negative and null effects on plant growth regardless of biochar type, dosage, or application method

Application of biochar resulted in null or negative biomass responses in ryegrass and clover regardless of biochar type, dosage, or application method. The average of the three mean responses of all treatments using MFT, SB, and MB revealed decreased ryegrass aboveground biomass relative to controls by 25%, leaf area by 17%, and belowground biomass by 35% (Fig. 1; Table 3). All three individual biochar types negatively impacted all growth traits examined in ryegrass (Fig. 1), but had no significant effect on clover growth traits (Table 3). Nodulation in clover did not differ across treatments (Table S2). All performance traits in ryegrass, but not clover, were inhibited by biochar at all dosages and this inhibition decreased with biochar addition rate. Linear regressions detected significant negative relationships of above- and below-ground biomass and leaf area with increasing biochar addition rate for all three biochars used in almost all comparisons (Fig. 2). The mean of all the biochar treatments at 50 t/ha showed reduced ryegrass aboveground

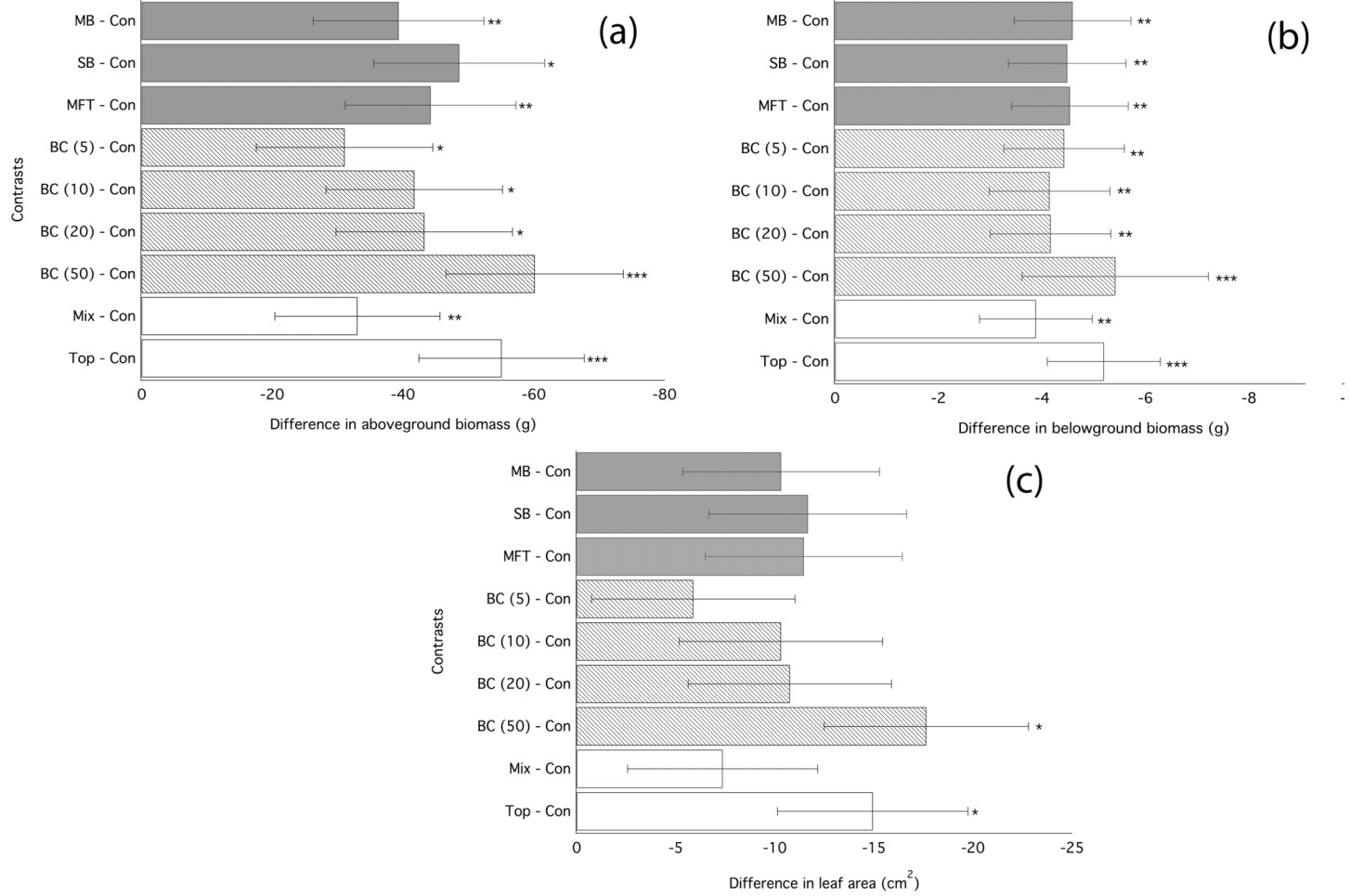

**Figure 1  Ryegrass growth responses to fresh biochars: contrasts between treatments.** Simultaneous confidence intervals (95%) for aboveground biomass (A), belowground biomass (B), leaf area (C) of ryegrass to biochar type (MB, SB, MFT), addition rate (5, 10, and 20 t/ha), and application method (Mixed or Top dressing (Top)) (Experiment 1). Significant differences between treatment means are denoted with asterisks ($P = < 0.05^*$, $< 0.01^{**}$, $0.001^{***}$).

biomass by 34%, leaf area by 28%, and belowground biomass by 43% (Fig. 1; Table 3). Mixing biochar directly into the soil was significantly less inhibiting than direct surface applications for all growth parameters in ryegrass (Fig. 1; Table 3).

## Experiment 2: heating and leaching biochar prior to application alleviates growth inhibition in ryegrass

Ryegrass grown in biochar either water leached or convection heated prior to application performed significantly better than ryegrass grown in un-processed biochar (Fig. 3). Un-processed biochar reduced ryegrass aboveground biomass by 24% (Fig. 3A), leaf area by 22%, and belowground biomass by 28% (Fig. 3B; Table 4). There was no difference between mixed and top-dressed applications of the un-processed biochar. Post-treating the biochar by water leaching for 24 h or heat treatment $\geq 100\,°C$ significantly increased aboveground biomass as compared to un-processed biochar (Fig. 3A). Biochar that was water-leached for 24 h increased aboveground biomass by 34%, and belowground

**Table 3 Mean growth responses values for clover and ryegrass.** Mean values for aboveground bio­mass (g), belowground biomass (g), and leaf area (cm$^2$) for ryegrass and clover grown in treatments with maple and spruce sawdust biochar made in an 80 L batch pyrolyzer with highest treatment temperature between 350–450 °C, and in treatments with maple/birch sawdust biochar produced in a flow-through screw fed pyrolyzer with highest treatment temperature of 500–600 °C (Experiment 1). "BC (dosage)" is the mean value for all treatments of biochar at that dose, and "BC mixed/top-dressing" is the mean values for all biochar treatments applied as mixed and top-dressings.

| Treatment | Aboveground biomass (g) | | Belowground biomass (g) | | Leaf area (cm$^2$) | |
|---|---|---|---|---|---|---|
| | Mean | Std. error | Mean | Std. error | Mean | Std. error |
| *Ryegrass* | | | | | | |
| 1. Control | 175.0 | 7.89 | 12.8 | 0.89 | 63.02 | 3.07 |
| 2. Maple batch BC | 136.2 | 3.45 | 8.21 | 0.27 | 52.70 | 1.29 |
| 3. Spruce batch BC | 126.9 | 6.29 | 8.32 | 0.61 | 51.40 | 2.4 |
| 4. Maple flow through BC | 131.3 | 4.76 | 8.32 | 0.30 | 51.55 | 1.63 |
| 5. BC 5 t/ha | 144.5 | 43.53 | 8.38 | 3.08 | 57.13 | 17.08 |
| 6. BC 10 t/ha | 133.8 | 44.80 | 8.65 | 3.34 | 52.71 | 16.23 |
| 7. BC 20 t/ha | 134.76 | 39.00 | 8.6 | 3.11 | 53.16 | 11.65 |
| 8. BC 50 t/ha | 114.93 | 41.39 | 7.33 | 3.62 | 45.23 | 14.54 |
| 9. BC mixed | 142.5 | 5.50 | 8.92 | 0.42 | 55.65 | 1.92 |
| 10. BC top-dressing | 120.32 | 5.36 | 7.59 | 0.43 | 48.06 | 1.91 |
| *Clover* | | | | | | |
| 1. Control | 7.97 | 1.03 | 1.98 | 0.45 | 13.66 | 2.91 |
| 2. Maple batch BC | 5.29 | 0.40 | 1.58 | 0.13 | 11.11 | 0.88 |
| 3. Spruce batch BC | 5.07 | 0.62 | 1.65 | 0.19 | 11.80 | 1.32 |
| 4. Maple flow through BC | 4.72 | 0.46 | 1.57 | 0.11 | 11.05 | 0.78 |
| 5. BC 5 t/ha | 5.15 | 0.51 | 1.63 | 0.14 | 11.27 | 1.09 |
| 6. BC 10 t/ha | 5.89 | 0.62 | 1.85 | 0.18 | 13.4 | 1.19 |
| 7. BC 20 t/ha | 5.15 | 0.68 | 1.85 | 0.19 | 12.9 | 1.24 |
| 8. BC 50 t/ha | 3.40 | 0.41 | 1.07 | 0.10 | 7.84 | 0.89 |
| 9. BC mixed | 4.51 | 0.34 | 1.54 | 0.10 | 10.62 | 0.733 |
| 10. BC top-dressing | 5.54 | 0.46 | 1.66 | 0.13 | 12.06 | 0.91 |

biomass by 22% compared to un-processed biochar (Table 4). Convection heating of biochar for 24 h at 100 and 150 °C increased leaf area by 20 and 23%, respectively (Fig. 3; Table 4). However, aboveground biomass and leaf-area of ryegrass grown with any processed biochars did not differ significantly from ryegrass grown in control (without biochar) soil (Table 4).

The water-based filtrate from biochar leaching applied immediately following transplantation significantly decreased growth of ryegrass as compared to plants grown in control soil (Table 4). Filtrate applied from the 24 h leaching most strongly inhibited growth, decreasing aboveground biomass by 40%, leaf area by 27%, and belowground biomass by 36% (Fig. 4; Table 4). Filtrate from the 30-min water leaching was less inhibiting, decreasing leaf area by 18%. The significant decrease in ryegrass growth traits due to the application of filtrate was similar in magnitude to the application of un-processed biochar (Fig. 4; Table 3).

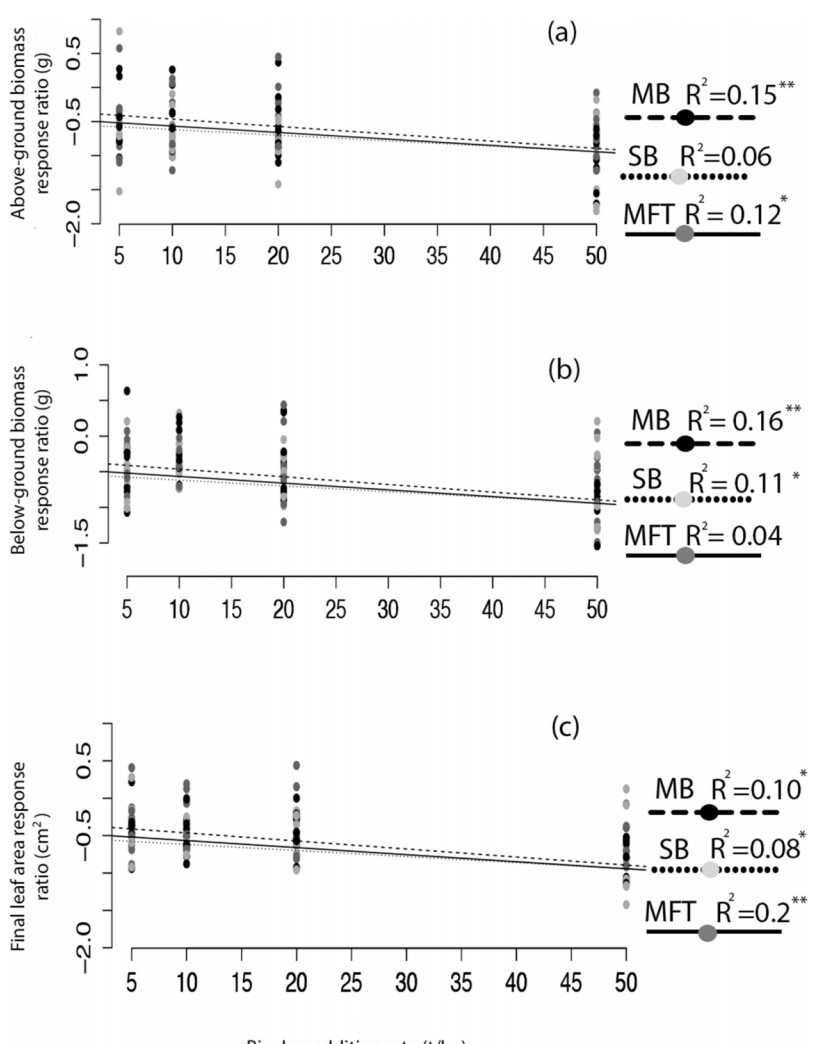

**Figure 2 Ryegrass growth response effect size as a function of biochar addition rate.** The relative effect size (response ratio–RR) of biochar treatment for aboveground biomass (A), belowground biomass (B), and leaf area (C) of ryegrass, as a function of biochar addition rate. Effect size is calculated as RR = ln(B/C), where B is mean response of the biochar treated plants, and C is the mean response of the non-biochar treated control plants. Regressions were performed for each biochar type (MB, SB, MFT); significant linear relationships are denoted with asterisks ($P = < 0.05^*, < 0.01^{**}, 0.001^{***}$).

## Mobile organic compounds detected with SPME-GC-MS

SPME-GC-MS of solid chars and leachates revealed primarily short-chained carboxylic acids, phenols, hydrocarbons; with notable differences between chars (Fig. 5). HS-SPME of MB qualified acetone, acetonitrile, acetic acid, 2-ethylbutyric acid, and butyric acid (Fig. 5A). HS-SPME of SB revealed acetic acid, benzene, decane, methyl isocyanide, ethylbenzene, dodecane, and 6-methyoxybenzofuran (Fig. 5B). DI-SPME of MB leachates revealed acetic acid, butyric acid, valeric acid, phenol, 2,4-di-tert-butylphenol and benzoic acid (Fig. 5C). DI-SPME of SB leachates similarly detected acetic acid, butyric acid, 2,4-di-tert-butylphenol and benzoic acid; however, oxime, and 5-methyoxybenzofuran were also detected (Fig. 5D).

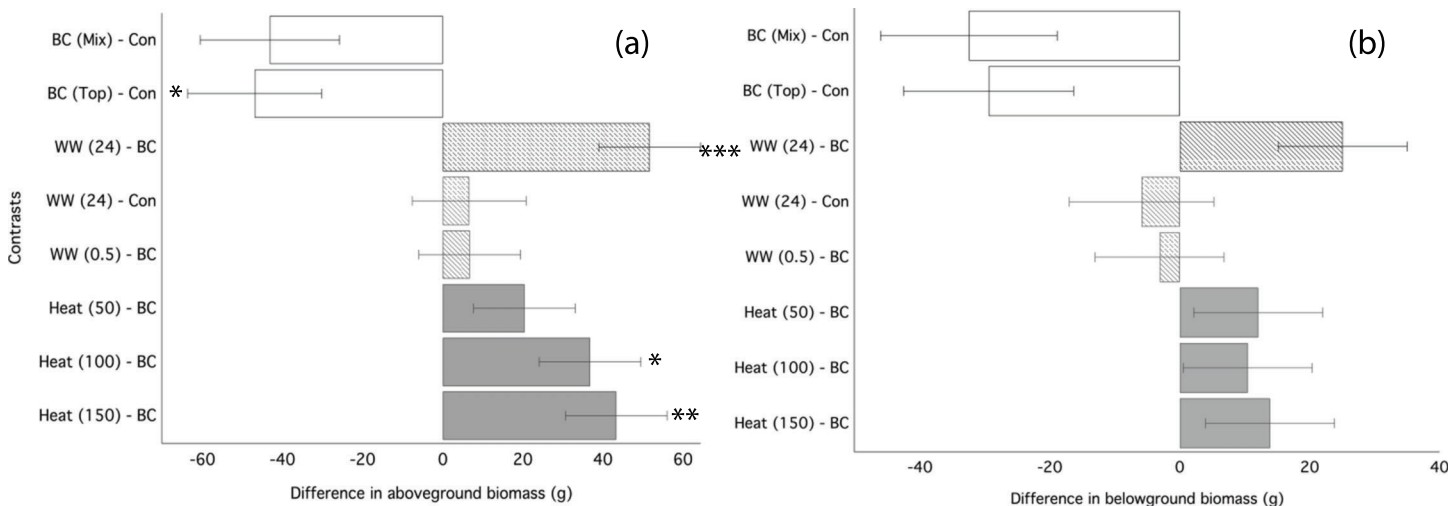

**Figure 3 Ryegrass growth responses to thermally treated and leached chars: contrasts between treatments.** Simultaneous confidence intervals (95%) for ryegrass aboveground biomass (A) and belowground biomass (B) responses to water washed (striped bars) and convection heated (grey bars) biochar (Experiment 2). BC, biochar (unwashed); BC (Mix), biochar, mixed; BC (Top), biochar, top-dressing. Abbreviations for heating (Heat) and washing (WW) treatments are defined in the methods. Significant differences between treatment means are denoted with asterisks ($P = < 0.05^*, < 0.01^{**}, 0.001^{***}$).

**Table 4 Mean values for plant growth responses to treated biochar.** Mean values for aboveground biomass (g), belowground biomass (g), and leaf area ($cm^2$) for ryegrass grown in 10 t/ha of fresh or treated (convection heated or water washed) maple/birch sawdust biochar produced in a flow-through screw fed pyrolyzer with highest treatment temperature of 500–600 °C (MFT), and values for ryegrass grown with 10 mL of "leachate" from water-washing of fresh biochars (Experiment 2).

| Treatments | Aboveground biomass (g) | | Belowground biomass (g) | | Leaf area ($cm^2$) | |
| --- | --- | --- | --- | --- | --- | --- |
| | Mean | Std. error | Mean | Std. error | Mean | Std. error |
| 1. Control | 195.50 | 10.59 | 122.20 | 12.57 | 63.74 | 2.50 |
| 2. Biochar | 150.27 | 9.07 | 91.33 | 6.45 | 47.65 | 2.36 |
| 3. Water washed 30 min | 157.13 | 11.13 | 88.13 | 7.37 | 49.10 | 2.07 |
| 4. Water washed 24 h | 202.13 | 9.46 | 116.31 | 8.84 | 59.19 | 1.90 |
| 5. Heated 50 °C | 170.81 | 9.24 | 103.31 | 5.50 | 53.72 | 2.36 |
| 6. Heated 100 °C | 187.18 | 1.45 | 101.69 | 1.02 | 57.84 | 0.31 |
| 7. Heated 150 °C | 193.75 | 1.48 | 105.12 | 0.80 | 57.20 | 0.49 |
| 8. Leachate 30 min | 141.75 | 7.24 | 98.88 | 7.37 | 56.69 | 4.37 |
| 9. Leachate 24 h | 116.38 | 5.35 | 78.38 | 7.48 | 46.53 | 1.50 |

## DISCUSSION

The negative effects of three sawdust biochars on ryegrass growth and the null response of clover growth observed in these experiments are contradictory to average plant growth trends described in recent meta-analyses (*Jeffery et al., 2011*; *Liu et al., 2013*; *Biederman & Harpole, 2013*; *Thomas & Gale, 2015*); however, these results are similar to the null to negative responses noted qualitatively in ~20% of prior studies (*Spokas et al., 2012*). High dosage (50 t/ha) and direct soil-surface applications of biochar (as opposed to mixing into the soil) were most inhibitory. Similar to *Biederman & Harpole (2013)*, who report

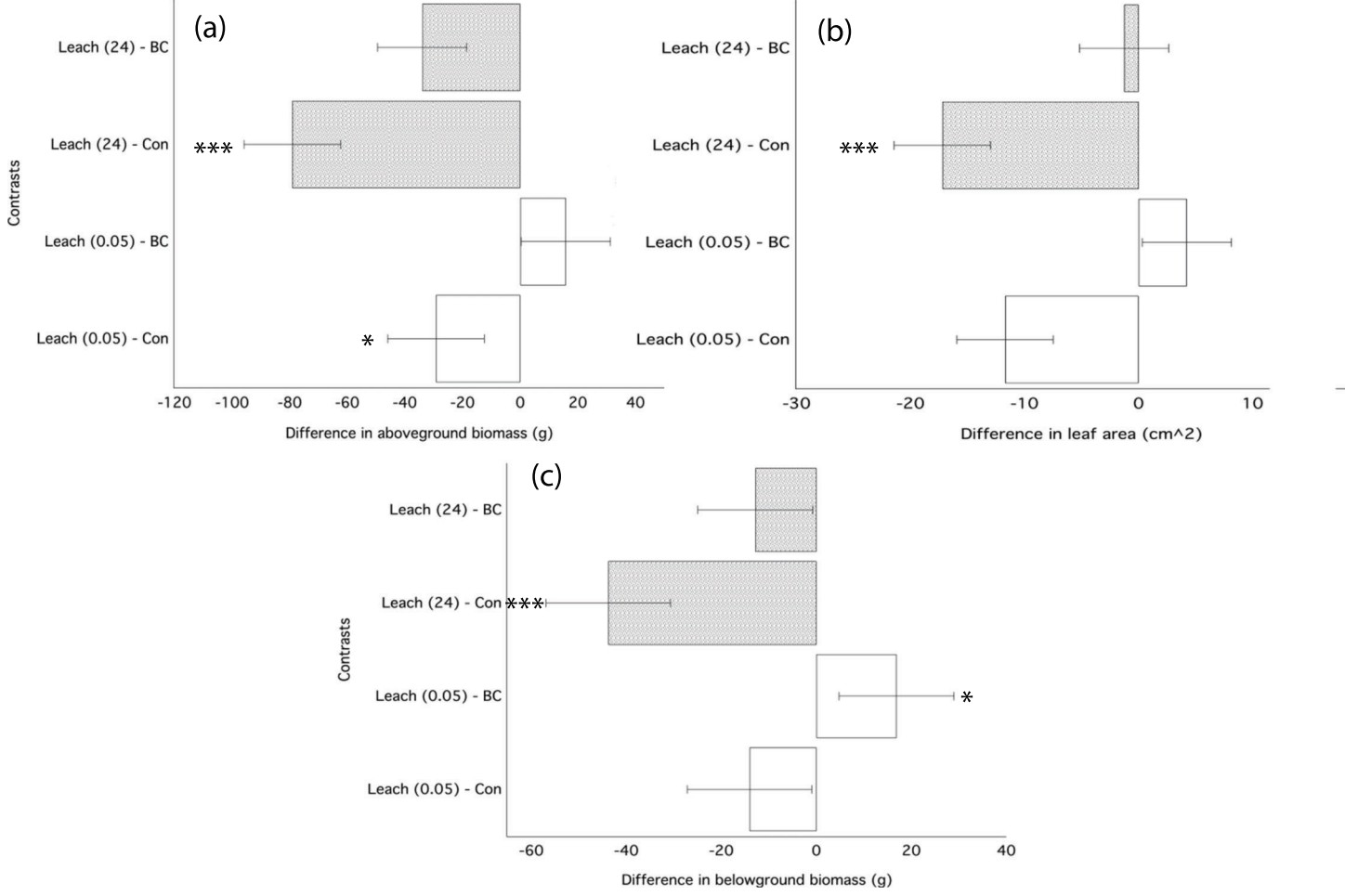

**Figure 4 Ryegrass growth responses to leachates from water washed chars.** Simultaneous confidence intervals (95%) for ryegrass aboveground biomass (A), belowground biomass (B), and leaf area (C) response in Experiment 2 to dissolved compounds from water-washed biochar leachates (Leach) applied as a 5 mL single dose following germination. Leachate treatments are defined in the methods. Significant differences between treatment means are denoted with asterisks ($P = < 0.05^*, < 0.01^{**}, 0.001^{***}$).

negative (but not significant) biomass response slopes to increasing biochar additions in almost half of 20 studies of plant responses to biochar in their meta-analysis, we detect a strongly negative linear relationship of performance traits to increasing biochar additions in ryegrass. Convection heating and water-based leaching of biochar prior to application alleviated growth inhibition of ryegrass. SPME-GC-MS suggests a number of common compounds which are volatile and leachable, and which we infer are generated by the pyrolysis process and re-condensed on or trapped within biochar (similar to *Hale et al., 2012*; *Spokas, 2013*; *Buss & Mašek, 2014*; *Kołtowski & Oleszczuk, 2015*; *Domene et al., 2015*; *Buss et al., 2015*; *Lievens et al., 2015*). The inhibition of ryegrass growth resulting from application of biochar leachate generated by water leaching was similar to that of un-processed biochar applied to soils (Table 4), suggesting the water soluble organic compounds qualified are very likely responsible for the growth inhibition observed.

Large differences in growth responses to biochar were observed between ryegrass (negative) and clover (null) to the three biochars applied (Table 3). This result is consistent

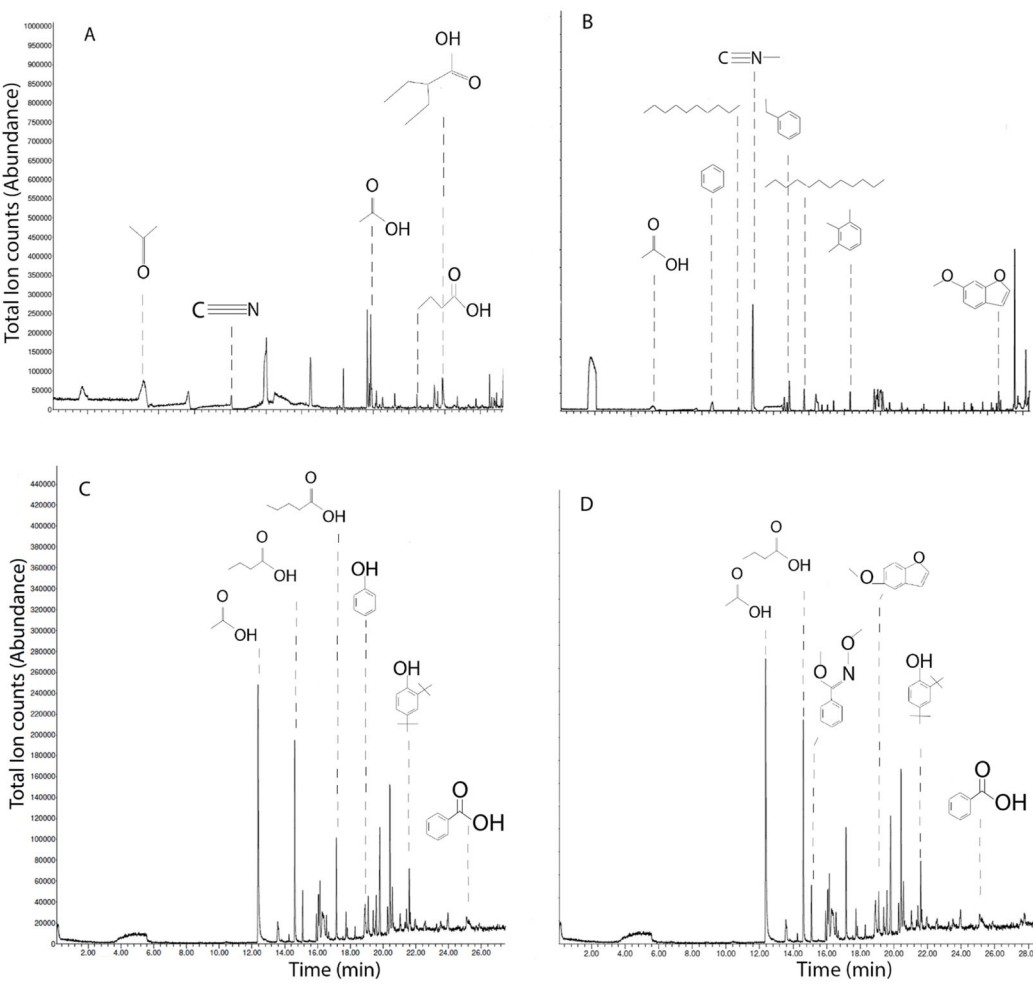

**Figure 5 Ion chromatograms of SPME on biochars and leachates.** Total ion chromatograms of HS-SPME of maple (A) and spruce (B) biochar, and DI-SPME of maple (C) and spruce (D) water leachates.

with a nitrogen immobilization mechanism (*Glaser, Lehmann & Zech, 2002*; *Asai et al., 2009*; *Deenik et al., 2010*), to which clover, as a legume, is more resistant. Nitrogen immobilization from biochar has been observed in temperate soils (see *Kloss et al., 2014*; *Rajkovich et al., 2012*), and can further be depleted by microorganisms that readily use other biochar derived leachates, such as high volatile matter and labile carbon (*Deenik et al., 2010*; *Marks, Alcañiz & Domene, 2014*). However, recent published work using two of these biochars (MB and SB) at the field site where our experimental soils were obtained (Haliburton, ON), showed that biochar caused no change to mineralized nitrogen, when applied at 5 t/ha as a top-dressing in a temperate forest stand (*Sackett et al., 2015*).

We suggest that species-specific phytotoxicity responses to volatile and leachable compounds from fresh biochar at least partially explains null and negative growth responses to biochar addition, and also contributes to the variation in plant responses to biochar. Several studies have now demonstrated phytotoxicity to fresh biochar, but there is little information regarding the source and chemistry of toxics involved (*Buss & Mašek, 2014*;

*Kołtowski & Oleszczuk, 2015*). *Hajaligol, Waymack & Kellogg (2001)* show rapid thermal decomposition of cellulosic materials at low to moderate (< 300 °C) pyrolysis temperatures forming a primary char that is capable of a second chemical transformation at moderate to high temperatures (> 400 °C). This first decomposition typically produces aldehydes, alcohols, and ketones, while the secondary transformation is likely to produce hydrocarbons and aromatics (*Hajaligol, Waymack & Kellogg, 2001*). We found evidence of both decompositions from moderate temperature pyrolysis of woody feedstocks used to produce the biochars in our study. Low to moderate temperature pyrolysis of woody-feedstocks are thus likely to produce a high concentration of primary decomposition derived low molecular weight compounds due to the high cellulose and lignin concentrations. Therefore, careful selection of feedstocks is necessary when creating biochars since trade-offs for restoration goals like high sorptive properties vs potential phytotoxicity are likely.

We suggest two causes for contamination of biochar from these mobile compounds: First, that pyrolysis retention times for biochar production is too short for sufficient evolution and diffusion through the pyrolysis system without being sorbed by biochar or trapped within its porous structure. This is especially likely in fast pyrolysis, although "slow" pyrolysis retention times of 0.5–2 h (or less) may also be too short (such as the biochar used here or in *Novak et al. (2014)*). The second cause for contamination of biochar may be due to re-condensation of volatiles during pyrolysis either from poor insulation or blockage in pyrolysis systems, and/or can occur as biochar cools (*Spokas et al., 2011*; *Buss & Mašek, 2014*). We suspect that one or both of these issues is common since mobile and volatile compounds are discovered in most biochars (*Spokas et al., 2011*), and since null and neutral plant responses are widely observed to biochar (*Spokas et al., 2012*). Much further investigation is necessary to resolve contamination of biochars, and particular focus should be given to identifying causal pyrolysis practices (e.g. retention times, cooling rates, flow rates, temperature uniformity, etc.).

The large variation in plant responses to biochar between tropical, temperate, and boreal systems may be partially due to phytotoxicity. Low growth responses to biochar in temperate systems in particular may be further hindered by nutrient immobilization by biochar (*Kloss et al., 2014*). We speculate that phytotoxicity responses in tropical systems treated with biochar may be reduced due to high soil turnover, rainfall, and/or microbial activity, partially explaining comparatively higher growth responses in these systems. In boreal systems, with low soil turnover and frequent fires, plants have likely adapted to mobile organic compounds associated with biochar (*Jeffery et al., 2011*; *Thomas & Gale, 2015*).

Convection heating and water leaching biochars mitigated the negative effects of applying biochar to a cambisolic soil (Table 4). Recent studies demonstrate similar effectiveness of leaching and thermal treatment to alleviate inhibition from hardwood, corn, and switchgrass biochars in *Rogovska et al. (2012)*, and in elephant grass and willow biochars in *Kołtowski & Oleszczuk (2015)*. We used high lignin-cellulose wood biochars produced at moderate pyrolysis temperatures (400–545 °C) at which peak volatilization occurs, likely producing high concentrations of volatile and leachable compounds found here: especially re-condensed bio-oils with a significant aqueous fraction containing

phenols and carboxylic acids (*Yu et al., 2007*; *Hale et al., 2012*). Higher temperature pyrolysis may generate similar and even greater concentrations of VOCs but may be less prone to re-condensation since cold-spot temperatures are possibly above condensation point.

Several prior studies have degassed (at > 100 °C) and water-leached compounds (including ethylene, acetone, benzene, propanol, K, Na, Mg, Zn, etc.) from similar wood biochars (*Wu et al., 2011*; *Novak et al., 2009*; *Spokas et al., 2011*). Indeed, low molecular weight compounds produced during natural fires are volatilized or leached relatively quickly from natural charcoals (*Ice, Neary & Adams, 2004*), which may contribute to rapid regeneration following fires (*DeLuca, MacKenzie & Gundale, 2009*). *Gundale & DeLuca (2007)* found positive growth responses in *K. macrantha* when wildfire-produced Douglas Fir and ponderosa pine biochar was applied, in contrast to growth suppression in un-processed biochar, suggesting weathering of chars alleviates phytotoxic effects of volatile and leachable compounds. It is thus possible that longer heating and leaching times prior to application might have resulted in positive growth responses to biochar in our experiment. Additionally, our biochar was stored in a sealed container prior to the experiment. Strong ventilation or aeration of chars while in storage may have prevented phytotoxicity.

## Leached compounds from biochar inhibit plant growth

Effluent from biochars leached with de-ionized water decreased ryegrass growth by up to 40% (Table 4), similar to the decreases in growth shown by treatments with the un-processed biochar. This result confirms the growth-inhibitory nature of the aqueous fraction of pyrolysis-produced bio-oils and is consistent with others that have found inhibitory effects from biochar-leached chemical compounds on plant germination (*Rogovska et al., 2012*; *Buss & Mašek, 2014*; *Kołtowski & Oleszczuk, 2015*) and microbial communities (*Lehmann et al., 2011*). Here we report inhibition in a variety of performance traits, rather than germination, from less intense leachings than in (*Rogovska et al., 2012*), and from more typical neutral to basic chars (pH = 6.5–7.4) than the acidic char studied by *Buss & Mašek (2014)* (pH = 3.64). Since we observed a positive relationship between leaching duration and compound concentration, we recommend that future applications of biochar use pyrolysis post-treatments that leach for longer durations. We watered our plants semi-daily to saturation, potentially leaching compounds quicker than would occur naturally. Phytotoxicity to biochar in areas of low rainfall, or otherwise under growth conditions with low watering frequency might indeed be greater.

Feedstock characteristics and the conditions of pyrolysis strongly influence the type and amount of mobile organic compounds from biochars (*Quilliam et al., 2013*; *Hale et al., 2012*), and subsequently, the toxicity of leachates is likely to also be related to system conditions, in particular soil characteristics and microbial composition. *Spokas et al. (2011)* report lower concentrations of volatile organic compounds in biochars produced from open-pit, kiln, slow, and steam activated pyrolysis compared to those produced via gasification and fast pyrolysis. Mobile organic compounds in aqueous leachates from poultry litter, but not corn stalk biochars, resulted in germination inhibition of cress seeds (*Lepidium sativum*) in *Rombolà et al. (2015)*. Indeed, spruce and maple charcoals had different volatile matter

content and compositions of mobile organic compounds in our study. The effects of mobile organic compounds on soil microorganisms will further influence plant responses (*Dutta et al., 2016*), with recent research showing shifts in microbial communities from biochar are influenced by soil properties (*Noyce et al., 2015*). Soil physical and chemical properties such as porosity, aeration, leachability, moisture content, and pH are also likely to play a critical role in determining hormetic effects of leached mobile organic compounds from biochar, and should be the focus of much further study.

## Qualification of phytotoxic compounds and the potential for hormesis

The increasingly reported null and neutral responses to biochar necessitates quick and reliable analysis of the compounds responsible. SPME-GC-MS promises to serve this role and is becoming the standard for qualification and even semi-quantification of volatile and leachable organic compounds in biochar (*Spokas et al., 2011*; *Buss et al., 2015*; *Rombolà et al., 2015*). We detected similar low molecular weight compounds to *Rombolà et al. (2015)* and *Buss et al. (2015)* with slight differences in GC-MS phases and procedures. However, we were unable to analyze the other maple biochar produced in a different pyrolysis unit and slightly different feedstock composition for mobile compounds, which potentially limits the array of compounds detected, as well as inference on the influence of pyrolysis conditions on the formation of mobile organic compounds. *Buss et al. (2015)* report concentrations of phenols and organic acids from moderate temperature pyrolysis of softwood chars in concentrations slightly lower ($\sim 22.4$ mgL$^{-1}$ and $\sim 52.2$ mgL$^{-1}$, respectively) than concentrations that have inhibited root elongation in prior studies (*Lynch, 1980*; *Feng et al., 1996*).

Pyroligneous acids, or "wood vinegars," have promoted plant growth and establishment in several studies (*Kadota, Hirano & Imizu, 2002*; *Du et al., 1998*). Acetic acid is the primary component of the chemical fraction in wood vinegars which have a long history of use as plant growth stimulators (*Mu, Uehara & Furuno, 2004*; *Mungkunkamchao et al., 2013*). *Mu, Uehara & Furuno (2003)* showed strong hormetic responses in seed germination of four species: *Lactuca sativa, Rorippa nasturtium-aquaticum, Cryptotaenia japonica, Chrysanthemum corronarium*. Strong inhibition of germination and radical growth was observed at high concentrations, while dilutions stimulated germination and growth, with obvious species-specific responses. Root development was increased, but not shoot development, when diluted wood vinegar was added to *Pyrus pyrifolia* cuttings (*Kadota, Hirano & Imizu, 2002*). Similarly, radicle length was increased when aqueous extracts of one of several charcoals was added to corn seeds in *Rogovska et al. (2012)*. Accurate quantification of volatile and leachable organic compounds in biochar is necessary to determine the type of hormetic plant responses; however, current SPME-GC-MS techniques offer only semi-quantification with noted difficulties (*Spokas et al., 2011*; *Dutta et al., 2016*). Germination tests could be utilized to evaluate hormetic effects from mobile organic compounds in biochar: principally, to identify compounds responsible, possible synergistic effects, and species-specific toxicity thresholds.

## CONCLUSIONS

In summary, the negative and null responses of two common forage crop species, ryegrass and clover, to biochar additions were contrary to the overall trend of positive plant growth responses presented in recent meta-analyses (*Jeffery et al., 2011*; *Liu et al., 2013*; *Biederman & Harpole, 2013*; *Thomas & Gale, 2015*). Our results strongly suggest that mobile organic compounds from biochar were responsible for this growth inhibition (primarily organic acids and phenols), as heating and leaching biochar before application alleviated this negative response, and addition of leachates alone replicated the negative responses observed. Chemical analysis of chars and leachates suggest possible specific molecules involved, including acetic acid, butyric acid, 2,4-di-tert-butylphenol and benzoic acid.

Our results suggest that negative responses to mobile compounds (in plants and soil biota) leached or evolved from biochars are an important mechanism accounting in part for the wide variation in plant growth responses to biochar described in the literature. Progressive volatilization and leaching almost certainly occurs in the field through weathering, both for charcoal produced during forest fires, and biochar applied as a soil amendment. Weathering of biochar will likely lead to rapid reductions in toxicity under field conditions. Additionally, soil flora and fauna likely metabolize mobile organic compounds rapidly further reducing toxicity in field soils with high soil floral and faunal diversity and abundance. Whether pre-weathering of biochars before application is beneficial (in terms of cost and reduction of negative effects on biota), or if natural weathering is sufficient to quickly remove toxic compounds, are important considerations for future studies and applied use of biochars. Investigations into production and storage methods that remove or decrease concentrations of compounds–such as heating and retention times, and ventilation, respectively–are needed. Efforts to identify the source of contamination of biochar, to accurately quantify toxic volatile and leachable compounds from biochars, and to test their potential toxicity to biota in a variety of systems and applications (i.e. temperate/tropical agricultural, forestry, environmental remediation, etc.) are also fundamental to optimize the use of chars as a soil amendment.

## ACKNOWLEDGEMENTS

We thank Haliburton Forest and Wildlife Reserve Ltd, Haliburton, ON for research support and provision of biochars and soil. Special thanks are also due to Chihiro Ikeda, Sossina Gezahegn, Aruna Kumari, Kathleen Manson, Janise Herridge, Carolyn Winsborough, Gowthaman Rajakumar and Katie Ungard for assistance with the experiments, and to Tony Adamo and Robert T. Gale for help with analytical work.

### Funding

Funding was provided by the Natural Sciences and Engineering Research Council of Canada. The funders had no role in study design, data collection and analysis, decision to publish, or preparation of the manuscript.

![PeerJ]

## Competing Interests

The authors declare that they have no competing interests.

## Author Contributions

- Nigel V. Gale conceived and designed the experiments, performed the experiments, analyzed the data, wrote the paper, prepared figures and/or tables, reviewed drafts of the paper.
- Tara E. Sackett conceived and designed the experiments, performed the experiments, wrote the paper, reviewed drafts of the paper.
- Sean C. Thomas conceived and designed the experiments, analyzed the data, contributed reagents/materials/analysis tools, wrote the paper, reviewed drafts of the paper.

## Data Deposition

The raw data has been supplied as Supplemental Dataset Files.

## Supplemental Information

Supplemental information for this article can be found online at http://dx.doi.org/10.7717/peerj.2385#supplemental-information.

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
