# Peer review of "Thermal treatment and leaching of biochar alleviates plant growth inhibition from mobile organic compounds"

_PeerJ, doi:10.7717/peerj.2385_

## Round 0.1 · original submission · Major Revisions

I consider your work interesting and with novel results that deserves publication. However you will need to improve the clarity of the experiments performed as well as the results section. Both reviewers have included different aspects of the manuscript that should be modified to increase the clarity of results and make your arguments and conclusions stronger. I will suggest taking most of them under your consideration and try to address all reviewers' suggestions or requirements. Also in the experiment in which you included different doses of biochar would be wise to perform a trend analysis to test if you found a significant effect of biochar doses, as should be done for a quantitative variable. It seems that for some parameters there is a significant increase of the measured variable as the biochar doses increase.

Reviewer 1 ·

Basic reporting

The paper is reasonably well written and includes an introduction based on a large number of previous and recent studies to demonstrate how the work fits into the broader field of knowledge. The authors provide a rationale in the Introduction by showing that a problem exists about the phytotoxicity of some biochars, and that there is little information regarding the source and chemistry of toxics involved.
However, the quality of the paper decreases because of the results, figures and tables (not well labeled and described) that are sometimes difficult to understand.

Experimental design

The aim of the paper is clear and the authors define enough well as to how the study contributes to filling the knowledge gap about the potential cause for null and negative plant of biochar on species. In particular, the authors applied five dosages of the biochars (0, 5, 10, 20, 50 t/ha) to a soil as direct surface applications and as complete soil mixes. Moreover, the authors examined the effects of leaching and heating treatments on biochar and studied the mobile organic compounds by SPME-GC-MS to identify candidate molecules involved in toxicity responses. However, the authors do not describe the methods with sufficient information to be reproducible by another investigator.

Validity of the findings

The results of experiment 1 and 2 are unclear and it is difficult understanding whether the conclusions are appropriately stated and the analysis of the data was well executed.

Additional comments

In the present paper entitled “Thermal treatment and leaching biochar alleviates plant growth inhibition from mobile organic compounds”, Gale et al. tested the effects of three different biochars on two common forage crop species (Lolium multiflorum and Trifolium repens). Moreover, the authors studied the molecules involved in toxicity responses by SPME-GC-MS.
The paper is relatively interesting because now there are several studies demonstrating phytotoxicity to fresh biochar, but there is little information regarding the source and chemistry of toxics involved. However, the results are presented in a less understandable form and the discussion is partly weak. For example, the results of experiment 1 (negative and null effects on plant…) are average of three biochars? BC (line 321) is SB, MB, MFT or average of three biochar?
Moreover, I did not fully understand why you used only MFT biochar for experiment 2 (modulation of effects through leaching and heating treatments) and analyzed by SPME-GC-MS the SB and MB biochars to study the organic compounds in biochars and in the water leachates.
Therefore, I would like to see much more discussion, integration and rationalisation of the results before it is published.

Detailed comments:
The abbreviation BC normally refers to Black Carbon.
Please write GC-MS instead of GCMS.
Line 23: Please change “Microexraction” to “Microextraction”.
Line 23: Please first write gas chromatography-mass spectrometry and then abbreviate.
Line 83: Please change “bio-oils” to “bio-oil”.
Line 88: Please change “(PAH’s)” to “(PAHs)”.
Line 127: Please use abbreviation instead of biochar.
Line 135: In order to clarify the text, should be “(5, 10, 20, and 50 t/ha)” instead of “(0-50 t/ha)”.
Line 151: Please first write gas chromatography-mass spectrometry and then abbreviate.
Lines 166-171: “Additional, lolium multiflorum ….plants (Welsch-Pausch, MacLachlan & Umlauf, 2002)” this part could be eliminated, in my opinion.
Line 175: Please change “11.33-cm2” to “11.33 cm2”.
Line 176: Please change “21-cm” to “21 cm”.
Line 188: Check parenthesis.
Lines 197-198: I did not understand this sentence. In particular, I did not understand “130, 5, 48 kg/ha, respectively”. It should be rewritten.
Line 218: Please mL instead of ml.
Line 235: “was determined” should be “were determined”.
Line 238: “was quantified” should be “were quantified”.
Line 240: Please use abbreviation instead of char.
Line 240: “was obtained” should be “were obtained”.
Lines 262-263: Please change “(HS)-SPME of solid chars, 0.5 grams of char” to “(HS)-SPME of BCs, 0.5 grams of sample”.
Line 263: “2.0 ug/mL of 1 ppm” should be “2.0 mL of 1 ppm”.
Lines 274-276: “by spiking 3mL of dionized water leachates (following washing method used for MFT) with 2.5 mL of 1 ppm O-eugenol, 1.0 mL of 2M KH2PO4 buffer, and 2.5 mL of 2-ethyl butyric acid as internal standards”
Maybe better: “ on 3 mL of deionized water leachates (following washing method used for MFT) with 1.0 mL of 2M KH2PO4 buffer. This sample was spiked with 2.5 mL of 1 ppm O-eugenol and 2.5 mL of 2-ethyl butyric acid as internal standards”.
Line 270: Please change “PDSM” to “PDMS”.
Line 291: Check parenthesis.
Line 305: Check parenthesis.
Line 357: Please use abbreviation instead of solid chars.
Lines 358-359: Please change “HD-SPME” to “HS-SPME”.
Line 388: Check parenthesis.
Line 405: Please change “hyrdrocarbons” to “hydrocarbons”.
Line 409: Please change “ VOC’s” to "VOCs”.
Line 454: “by (Buss & Mašek, 2014) (2014)” should be “Buss & Mašek (2014)”.
Line 476: “(Mu, Uehara & Furuno, 2003) showed” should be “Mu, Uehara & Furuno (2003) showed”
Line 625: Check reference. 2014b? or 2014?
Line 658: Check reference.
Line 740: Check reference.
Line 767: Check reference.
Figures 1, 2 and 3: Do not well labeled and described. Therefore, they are difficult to understand.

Reviewer 2 ·

Basic reporting

The paper Paper fails on Soil Science area. Paper is focused on the toxicity of three different biochars in a greenhouse experiment and also the use of the same biochars after thermal treatment and leaching to study the toxicity of organic compounds.

The work is clear and the introduction is well focused. The structure is according to journal guidelines. Also, the tables and figures are of good quality.

Experimental design

Experimental Design is correct. Paper could be improve doing a toxicity test, as Zucconi test, to study the percentage of germination and the germination index. The test should be done with the treated soil and also with only biochars. This test can help to understand better the results.

With respect to biochars, authors should improve the characterisation of the biochar. For example, I recommend follow the guidelines of the International Biochar Initiative and also it will be very useful to include the proximate analysis (fixed carbon, volatile matter, ash content).

Also, authors should be explained better the choice of pyrolisis conditions. Indeed, could be pyrolysis temperature related with the composision of mobile compounds?

According to that, authors should improve the discussion about the properties of different biochars.

Validity of the findings

Authors conclude "Our results strong suggest that mobile organic compounds from BC were responsible for this growth inhibition (primarily organic acids and phenols), as heating and leaching BC before application alleviated this negative response and addition of leachates alone replicated the negative responses observed." and that results are contrary to "overall trend of positive plant growth responses presented in recent meta-analyses".

I am agree with the conclusions, but these conclusions are for these biochars and probably for other biochars or other soil the results will be different. Authors must support better this idea incluiding the type of biochar that has had negative or positive effect in plant growth in the cited papers.

Additional comments

See above

---

## Round 0.2 · Minor Revisions

I consider that you manuscript can be finally accepted if you are able to address comments made by reviewers concerning incorporation of some of the literature related to toxicity test in your discussion. Although I consider that including the determination of fixed carbon and volatile matter will increase better the characterization of your biochars since the other reviewers do not consider this experiments compulsory you do not need to provide this information

Reviewer 1 ·

Basic reporting

No Comments

Experimental design

No Comments

Validity of the findings

No Comments

Additional comments

The authors have adequately addressed my concerns with the original version of the manuscript. Therefore, I recommend the publication of this manuscript in PeerJ journal.

Reviewer 2 ·

Basic reporting

The paper Paper fails on Soil Science area. Paper is focused on the toxicity of three different biochars in a greenhouse experiment and also the use of the same biochars after thermal treatment and leaching to study the toxicity of organic compounds.

The work is clear and the introduction is well focused. The structure is according to journal guidelines. Also, the tables and figures are of good quality.

Experimental design

Experimental Design is correct. Paper could be improve doing a toxicity test

Validity of the findings

As the first time, I am agree with the conclusions, but these conclusions are for these biochars and probably for other biochars or other soil the results will be different. Authors must support better this idea including the type of biochar that has had negative or positive effect in plant growth in the cited papers

Additional comments

My comments about the basic reporting, experimental design, validity of findings are the same that in the first version.

With respect to the answer of the authors, they did not address all my comments,specially:

1. With respect to toxicity test, they refuted my suggestions with some papers about germination inhibition from similar mobile organic compounds leached from similar biochars in several species but they did not mentioned papers about positive effect on germination. There are a lot.

2. With respect to characterisation, they followed some indications but they did not determined the fixed carbon and volatile matter. These parameters are very important to understand the behaviour of biochar. I know that there are a lot of papers that do not include this parameters. This fact probably is due to the big amount of papers about biochar of the last years and probably, the reviewers do not are specialist in the topic.
Also, it is not dangerous to determine the volatile matter. You can consult different papers and you do not need to do in a muffle furnace. The proximate analysis is usually made by a TG analysis. Please consult the literature, it is vey easu to do and the analysis is not expensive.

In my opinion, paper should address these points before to be accepted.

---

## Round 0.3 · accepted · Accept

You have addressed all comments made by the reviewer and your manuscript has improved in clarity